# Involvement of PAR-2 in the Induction of Cell-Specific Matrix Metalloproteinase-2 by Activated Protein C in Cutaneous Wound Healing

**DOI:** 10.3390/ijms25010370

**Published:** 2023-12-27

**Authors:** Sohel M. Julovi, Kelly McKelvey, Nikita Minhas, Yee-Ka Agnes Chan, Meilang Xue, Christopher J. Jackson

**Affiliations:** 1Sutton Arthritis Research Laboratory, Kolling Institute of Medical Research, University of Sydney, Pacific Highway, St. Leonards, NSW 2065, Australia; kelly.mckelvey@sydney.edu.au (K.M.); minhas.nikita@gmail.com (N.M.); agnes.chan@sydney.edu.au (Y.-K.A.C.); meilang.xue@sydney.edu.au (M.X.); chris.jackson@sydney.edu.au (C.J.J.); 2Kidney Injury Group, Centre for Transplant and Renal Research, Westmead Institute for Medical Research, Westmead, NSW 2045, Australia

**Keywords:** activated protein C, cutaneous wound healing, murine, matrix metalloproteinase-2, -9, protease activator receptor-2

## Abstract

We previously reported that human keratinocytes express protease-activated receptor (PAR)-2 and play an important role in activated protein C (APC)-induced cutaneous wound healing. This study investigated the involvement of PAR-2 in the production of gelatinolytic matrix metalloproteinases (MMP)-2 and -9 by APC during cutaneous wound healing. Full-thickness excisional wounds were made on the dorsum of male C57BL/6 mice. Wounds were treated with APC on days 1, 2, and 3 post-wounding. Cultured neonatal foreskin keratinocytes were treated with APC with or without intact PAR-2 signalling to examine the effects on MMP-2 and MMP-9 production. Murine dermal fibroblasts from PAR-2 knock-out (KO) mice were also assessed. MMP-2 and -9 were measured via gelatin zymography, fluorometric assay, and immunohistochemistry. APC accelerated wound healing in WT mice, but had a negligible effect in PAR-2 KO mice. APC-stimulated murine cutaneous wound healing was associated with the differential and temporal production of MMP-2 and MMP-9, with the latter peaking on day 1 and the former on day 6. Inhibition of PAR-2 in human keratinocytes reduced APC-induced MMP-2 activity by 25~50%, but had little effect on MMP-9. Similarly, APC-induced MMP-2 activation was reduced by 40% in cultured dermal fibroblasts derived from PAR-2 KO mice. This study shows for the first time that PAR-2 is essential for APC-induced MMP-2 production. Considering the important role of MMP-2 in wound healing, this work helps explain the underlying mechanisms of action of APC to promote wound healing through PAR-2.

## 1. Introduction

Cutaneous wound healing requires a complex and dynamic array of partially overlapping phases: inflammation, angiogenesis, new granulation tissue formation, re-epithelialisation, and tissue remodelling [1,2]. Matrix metalloproteinases (MMPs), a class of extracellular matrix (ECM)-degrading proteinases, are secreted by different cell types such as keratinocytes, fibroblasts, and inflammatory and immune cells during wound healing. Among the MMPs, differential expression and activity of MMP-2 and MMP-9 are observed during wound healing at specific stages of wound repair [2,3].

During the initial inflammatory stage of wound healing, MMP-9 breaks down ECM components like collagen and elastin, and facilitates the clearance of damaged tissue, allowing immune cells to access the wound and initiate the healing process [4]. In the later phases, MMP-2 and -9 promote new granulation tissue by stimulating keratinocyte, fibroblast, and endothelial cell proliferation and migration [2,5,6,7]. Not surprisingly, skin wound healing is impaired in MMP-2- and MMP-9-deficient mice [8,9] and in double-knock out mice [10].

Activated protein C (APC), a physiological serine protease, promotes cutaneous wound healing [11,12], by stimulating keratinocyte proliferation, migration, barrier integrity, and inhibiting apoptosis [12,13]. APC initiates a series of events that modulate the balance between pro- and anti-apoptotic factors [13,14,15,16,17]. The inhibition of nuclear factor-kappa B (NF-κB) signalling by APC, as observed in studies such as that by Cheng et al. [18], represents another mechanism through which APC influences apoptosis and inflammatory responses [19]. The precursor, PC, binds to its homing receptor, endothelial protein C receptor (EPCR), and is activated by the cleavage of the N-terminal by thrombin bound to thrombomodulin; APC, still in association with EPCR, then stimulates protease-activated receptors (PARs) by the cleavage of the N-terminal to unmask a tethered ligand [3,14,20]. The PAR subfamily of G-protein-coupled receptors plays an important role in coagulation, inflammation, and tissue remodelling [18,20,21].

We and others reported that human keratinocytes and dermal fibroblasts express PAR-2 [20,22,23]. Activation of PAR-2 stimulates the release of inflammatory mediators such as interleukin (IL)-6, IL-8, prostaglandin E2, matrix metalloproteinase (MMP)-2, and MMP-9, suggesting an important role of PAR-2 in inflammation and tissue remodelling [24,25]. Previously, in a rat model, we reported that the broad-spectrum MMP inhibitor, GM6001, abolishes the ability of APC to promote wound healing [11]. Using knock-out mice, we have established the involvement of PAR-2 in APC-induced cutaneous wound-healing acceleration [22]. However, the entire mechanism of APC/PAR2-mediated promotion of cutaneous wound healing is unresolved. There is currently no study that addresses the role of PAR2 in MMP-2 or MMP-9 regulation in wound healing. In this study, we demonstrate that PAR2 is required for APC regulation of MMP-2 in murine skin wound healing and human skin keratinocytes.

## 2. Results

### 2.1. Activated Protein C Accelerates Murine Cutaneous Wound Healing

Histological analyses revealed that the 6 mm excised wounds had contracted to approximately 4 mm after 24 h, with eschar formation observed in both APC- and PBS-treated mice (Figure 1). On day 3, the wounds treated with PBS showed minimal changes, while those treated with APC exhibited hyperproliferation of the epidermis beneath the eschar. Additionally, the migrating epithelial edge had nearly closed the wound, leaving a gap of approximately 1 mm.

By day 6, the PBS-treated wounds displayed hyperproliferation of the epidermis and re-epithelialization beneath the persistent eschar. By contrast, the APC-treated wounds had shed the eschar, restructured the epithelial layer to its normal thickness, and were only distinguishable from the PBS-treated wounds by the disorganization of collagen in the dermal wound tract. This indicated near-complete healing of the skin wounds in the APC-treated group, and complete wound closure was reduced by ∼3 days in APC-treated mice [22].

We have previously shown that APC uses PAR-2 to promote murine cutaneous wound healing [22], and reported that the wound area was significantly higher (*p* < 0.01 and *p* < 0.004 on days 3 and 6, respectively; *n* = 6) in APC-treated PAR-2 KO mice compared to APC-treated WT mice [22]. Here we confirmed the role of PAR-2 in a mouse excisional wound model. Photomicrographs on days 3 and 6 showed that the wound area was wider in PAR-2 KO mice than in WT mice (Appendix A).

### 2.2. APC-Treated Wounds Induced Activity of MMP-2 and MMP-9 Earlier Than Control Wounds

MMP-2 and -9 expression and activation are known to be regulated by APC [12] and play fundamental roles in skin wound healing through the recruitment and clearance of immune cells, the degradation of damaged fibrillar collagen, the promotion of angiogenesis, and ECM remodelling [26]. The temporal profile of MMP-2 and MMP-9 expression during murine wound healing was measured using gelatin zymography (Figure 2A). Densitometric analysis showed that total (pro and active) MMP-9 expression was significantly increased at day 3 and gradually decreased from day 6 to day 13. By contrast, treatment with APC accelerated MMP-9 expression at day 1 (Figure 2B). The MMP-9 expression in APC-treated wounds gradually reduced from day 3 to day 13 and significantly reduced at day 6 compared to PBS-treated wounds (Figure 2B). The total MMP-2 in the wound homogenates increased over time, significantly peaking in APC-treated wounds at day 6 compared to PBS-treated wounds (Figure 2B).

MMP-9 expression was expressed by keratinocytes in the epithelial migrating edge, neutrophils, monocytes and fibroblasts in the wound dermis (Figure 3A and Appendix A). Enhanced expression of MMP-9 was observed at day 3 in APC-treated wounds, while it was observed at day 6 in PBS-treated wounds (Figure 3A). Evidence of non-cell-associated MMP-9 expression was also noted in the wound tract of PBS-treated wounds (Appendix A). MMP-2 expression by keratinocytes, fibroblasts, and endothelial cells at the wound margin and in the wound tract was minimal at day 1 in both PBS- and APC-treated wounds (Figure 3B). On day 6, APC led to an increase in MMP-2 expression in both the epidermis and dermis (Figure 3B and Appendix A), which is consistent with the Figure 2.

### 2.3. APC-Induced MMP-2 by Human Primary Keratinocytes Requires PAR-2

We have previously demonstrated that 1 μg/mL of APC stimulates human keratinocyte proliferation via transactivation of the PAR-2 and PI3K/Src/Akt pathways [22]. Little is known on the involvement of PAR-2 in APC-induced MMP-2 and MMP-9 activation in human keratinocytes. Neonatal keratinocytes were treated with increasing doses (0 to 5 μg/mL) of APC for 48 h and we found that APC activates MMP-2 at 1 µg/mL (Figure 4A) and is limited by αPAR-2 antibody levels (Figure 4B). Consistently, silencing of PAR-2 reduced the synthesis of total MMP-2 by 25–50% (Figure 4C). Interestingly, basal activation of MMP-9 was also inhibited by the αPAR-2 antibody, while APC had no effects on MMP-9 activation (Figure 4A,B).

### 2.4. PAR-2 Deficiency Reduces MMP-2 Activation by Murine Dermal Fibroblasts

Dermal fibroblasts play a critical role in the production of new granulation tissue during cutaneous wound healing. In the presence or absence of APC, fibroblasts produced only very faint bands for MMP-9, but strong bands for MMP-2 (Figure 5A). The basal level of pro-MMP-2 was significantly higher in PAR-2 KO mouse fibroblasts compared to WT mouse fibroblasts (Figure 5A,B). APC had no effect on pro-MMP2 in PAR-2 KO mice at any concentration, while it increased pro-MMP2 in WT mice at 10 μg/mL (Figure 5A,B). By contrast, the basal level of active MMP-2 was 30% lower in PAR-2 KO mouse fibroblasts compared to WT mouse fibroblasts, and APC dose-dependently increased active MMP-2 in both WT and PAR2 KO mice dermal fibroblasts (Figure 5A,C). However, MMP-2 activation by APC in PAR2-KO mouse dermal fibroblasts was reduced 2–5-fold when compared to WT mouse fibroblasts (*p* = 0.0002 to *p* < 0.0001, Figure 5C), suggesting that PAR-2 is the key regulator of APC-induced MMP-2 activation.

## 3. Discussion

An increasing body of evidence demonstrates that APC improves skin wound healing by reducing inflammation and promoting angiogenesis, granulation tissue formation, and re-epithelialisation in vitro [12], in vivo [11,22], and clinically [27,28]. We have demonstrated that APC improved wound healing in WT murine wounds, but not PAR-2 KO mice [22], indicating that APC acts through PAR-2. Almost all cell types present in the skin express PAR-2, including epidermal keratinocytes, dermal fibroblasts, endothelial cells, afferent neuron terminals [29,30], mast cells [31], neutrophils [32,33], and monocytes [34]. PAR-2 expression by at least some of these cells is increased following APC treatment [22].

PAR-2 is activated by proteolytic cleavage of its N-terminus by various serine proteases [35,36,37]. Once activated, PAR-2 initiates a cascade of intracellular events leading to diverse cellular responses [10,36,38]. One prominent downstream effect of PAR-2 activation is the modulation of MMP expression, particularly MMP-2 and MMP-9 [23,24]. The relationship between PAR-2 and MMP-2 and -9 production has been established in various physiological and pathological contexts [10,39]. For instance, in tissue remodelling and wound healing, PAR-2 activation has been implicated in the regulation of MMP-2 and MMP-9, facilitating the controlled degradation of extracellular matrix proteins to enable cell migration and tissue repair [23,39,40]. Additionally, PAR-2-mediated MMP-2 and MMP-9 expression has been shown to be crucial in processes such as angiogenesis [41], where the remodelling of blood vessels requires the coordinated action of proteases [42].

In vitro, APC dose-dependently increases MMP-2 activation, migration, invasion, and/or barrier integrity in cultured human primary keratinocytes [3,43], endothelial cells [44], and dermal fibroblasts [3,4,12]. The current study demonstrated that MMP-2 expression or activation in human cultured primary keratinocytes and murine dermal fibroblasts in response to APC was reduced when PAR-2 was blocked, silenced, or genetically inhibited. In agreement, we have previously shown that APC accelerates wound healing in WT mice; however, it has a negligible effect in PAR-2 KO mice [22].

Unlike the consistent temporal expression of PAR-2 in skin cells [20,22,23], an important finding in the current work is that MMP-2 and -9 expression during wound healing was not only divergent among cell types but, importantly, altered over time. Consistently with the literature [45], our IHC showed that MMP-9 was primarily produced by the leading edge of the epithelial tongue and peaked at days 1 and 3. By contrast, MMP-2 was expressed predominantly by epidermal keratinocytes, dermal fibroblasts, and endothelial cells with expression progressively increasing until complete wound healing. Quantitative assessment of gelatinase expression in the wounds confirmed APC’s early (day 1) vs. late (day 6) induction of MMP-2 vs. MMP-9 activity, respectively.

The reasons for the differential temporal expression of the two gelatinases during wound healing are not entirely clear; however, one leading explanation relates to their effect on inflammation. In the context of inflammation, PAR-2 activation has been linked to the induction of MMP-2 and MMP-9 expression, exacerbating tissue damage and inflammation [10,39]. Early MMP-9 activity would be beneficial during the initial inflammatory phase of healing, as it activates the pro-inflammatory cytokines IL-1β and IL-8, resulting in amplified leukocyte influx and increased inflammation [46]. Although we did not observe substantial APC-induced MMP-9 activity in human keratinocytes or mouse dermal fibroblasts, this has been observed in other tissues such as cartilage [47]. It is possible that APC-induced MMP-9 activation may be cell/tissue context-dependent. Alternatively, based on the early extracellular accumulation of MMP-9 we observed in mouse wounds, other local proteases or other factors may be at play. Interestingly, the level of active MMP-9 was reduced in APC-treated PAR-2-disrupted keratinocytes.

By contrast, MMP-2 likely plays a prominent role at the later stages of wound healing, not only by inducing cell migration and regulating the turnover of ECM, but also by dampening inflammation—an important step, as inflammation prevents healing if it persists for many days [3]. MMP-2 exerts anti-inflammatory effects via the truncation and inactivation of MCP-3, a CC chemokine that promotes leukocyte chemotaxis, which results not only in blocking the initiation of an in vivo inflammatory response but also in completely abrogating pre-existing inflammation [48].

A novel finding of this study is that PAR-2 is essential for APC-induced MMP-2 production but not for its activation, which helps explain the poor wound healing in PAR2 KO mice, in the presence or absence of APC [22]. Figure 6 outlines a proposed mechanism of action. Briefly, MMP-2 synthesis is achieved through PAR-2, although the direct cleavage of PAR2 by APC is yet to be described. An interesting mechanism described by Madhusudhan, Kerlin, and Isermann [49] may be involved, whereby APC cleaves PAR3 (or PAR1), which is bound to PAR-2 as a heterodimer, and trans-activates PAR2. Subsequent intracellular signalling pathways are implicated in the regulation of MMP-2 (and MMP-9) expression downstream of PAR-2 activation [35,37], including mitogen-activated protein kinase (MAPK) [36,38], nuclear factor-kappa B (NF-κB), and others [35,36,37]), which converge to modulate the transcriptional activity of genes encoding MMPs [36]. Finally, APC can directly activate MMP-2 independent of PAR-2 [44]. Understanding the exact mechanisms through which PAR-2 influences MMP-2 (and MMP-9) expression in cutaneous wound healing will assist in the development of targeted therapeutic interventions.

## 4. Materials and Methods

### 4.1. Animals

Studies were performed in male wild-type (WT), and PAR-2 KO mice. Mice were all in the C57BL/6J background and at 6 weeks of age when starting the wounding protocol. Mice were obtained from and housed at Kearns Facility, Kolling Institute of Medical Research, The University of Sydney, under a 12 h light/dark cycle with access to standard chow and water ad libitum as described previously [22]. All animal studies were performed in accordance with the Australian code for the care and use of animals for scientific purposes developed by the National Health and Medical Research Council (NHMRC).

### 4.2. Wound Healing Model

Animals were anaesthetised with isoflurane and oxygen. Full-thickness skin wounds extending through the *panniculus carnosus* were made with iris scissors, and a sterile 6 mm punch biopsy tool was used to outline a pattern on the dorsum of the mice as described before [22]. Recombinant APC (Xigris, Drotrecogin alfa (activated); Eli Lilly, Indianapolis, IN, USA) 10 µg/wound/day in a volume of 20 µL phosphate-buffered saline (PBS) or PBS alone (control) was topically applied to the wounds for 3 consecutive days. Animals were maintained under anaesthesia for 20 min to allow absorption of the solution. Wounds were left open, and animals were housed in individual cages. Wound healing was assessed by taking digital photographs and blindly measuring the wound area with a Visitrak digital device (Smith and Nephew, Macquarie Park, NSW, Australia) [22]. Mice were euthanised at indicated days and the area of skin around the healing wound was excised and processed blindly for further analysis.

### 4.3. Histology and Immunohistochemistry

Histology and immunohistochemistry were performed as previously described [22]. Briefly, wound tissue was fixed with 10% neutral-buffered formalin and paraffin sections (4 μm) stained with Mayer’s haematoxylin and eosin on randomly chosen samples. For immunohistochemistry, sections were incubated with rabbit anti-MMP-2 and anti-MMP-9 and isotyped-matched rabbit IgG (control; 1 µg/mL, Santa Cruz Biotechnology Inc., Santa Cruz, CA, USA). For immunodetection, a Dako EnVision+ System-HRP labelled polymer detection kit (Dako, Carpinteria, CA, USA) was used with ImmPACT NovaRED peroxidase (HRP) substrate (Vector Laboratories, Burlingame, CA, USA), and counterstained using Mayer’s hematoxylin and Scott’s bluing solution. After mounting, sections were observed under a light microscope (Eclipse Ci; Nikon, Tokyo, Japan) and micro-graphed using a DS-Fi1 CCD camera (Nikon, Tokyo, Japan). All samples were stained in a single assay to exclude between-run variability. The wound gap was measured by drawing a line the width of the wound, and the number of pixels was recorded using ImageJ. The pixels of the line were divided by the pixels of the scale bar as indicated in the figures.

### 4.4. Cell Culture and Reagents

Human keratinocytes were isolated from neonatal foreskins, as described previously [22], and cultured in keratinocyte serum-free medium (Invitrogen, Carlsbad, CA, USA). Dermal fibroblasts from WT and PAR-2 KO mice were isolated as described [50] and cultured in Dulbecco’s modified Eagle’s medium (DMEM) media (Sigma-Aldrich, St Louis, MO, USA). Both media were supplemented with 10% *v*/*v* heat-inactivated fetal bovine serum, and 100 U penicillin–streptomycin cells cultured in a humidified atmosphere at 37 °C containing 5% CO_2_. Primary cultured keratinocytes and murine dermal fibroblasts were seeded into 24-well culture plates and once confluent, treated with recombinant 0–10 µg/mL APC (Xigris; Eli Lilly, Indianapolis, IN, USA) for 24 h with or without pre-treatment of 10 µg/mL PAR-2blocking antibody (SAM11; Santa Cruz Biotechnology, Santa Cruz, CA, USA) for 2 h. Cultured supernatants were collected for the detection of MMP-2 and MMP-9. The use of human tissues was approved by the Northern Sydney Local Heath District Human Research Ethics Committee.

### 4.5. Knockdown of PAR-2 through siRNA

Expression of PAR-2 in primary keratinocytes was silenced with PAR-2 siRNA as described previously [22]. Briefly, cells were transfected with PAR-2 siRNA (5′-GGA AGA AGC CUU AUU GGU A-3′) or non-specific siRNA (control; 5′-GGC UAC GUC CAG GAG CGC ACC-3′; Ambion, Austin, TX, USA) using a siPORT NeoFX transfection reagent (Ambion, Austin, TX, USA) for 24 h. Transfected cells were then seeded into 24-well plates (4 × 10^5^ cells/mL) and treated with 1 µg/mL of recombinant APC for 48 h. Cells and culture supernatants were collected for the detection of protein expression. PAR-2 knockdown efficiency (>80%) was confirmed via immunoblot analysis [22].

### 4.6. Zymography

MMP-2 and MMP-9 protein secretion and activation were detected using gelatin zymography under non-reducing conditions as described previously [12]. Relative levels of MMP-2 and MMP-9 were semi-quantified using ImageJ software, V 1.8.0 (NIH, Bethesda, Rockville, MD, USA).

### 4.7. Fluorometric Assay

PAR-2 siRNA-transfected keratinocytes were treated with 1 μg/mL of APC for 48 h [29]. The conditioned media were assessed for MMP-2 activity using the SensoLyte Fluorometric MMP-2 assay (AnaSpec, Fremont, CA, USA) according to the manufacturer’s instructions. Briefly, MMP-2 within the cell supernatant was activated by the addition of 1 mM amino-phenyl mercuric acetate (APMA). Following the addition of the MMP-2 substrate, the fluorescent intensity was measured at indicated time points.

### 4.8. Statistical Analysis

Data were analysed using Student’s *t*-test (parametric data) or 1- or 2-way ANOVA for multiple-group (*n* > 2) comparisons. Sidak or Tukey multiple-comparison post-tests were performed where relevant. A *p*-value of <0.05 was assumed to be significant.

## Figures and Tables

**Figure 1 ijms-25-00370-f001:**
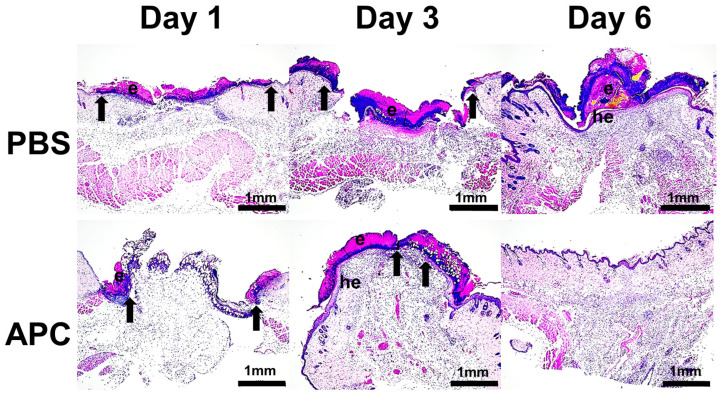
APC accelerates wound healing in C57/Bl6 mice. Full-thickness 6 mm diameter wounds were made and treated topically with 20 μL of PBS or APC (10 μg) once a day for 3 consecutive days. Haematoxylin and Eosin (H&E)-stained paraffin sections from day 1, 3, and 6 post-wounding. Sections represent 6 separate experiments in each individual group. Arrows denote the leading edge of the migrating epithelial tongue. e, eschar; he, hyper-proliferative epithelium. Scale bar = 1 mm. Abbreviations: APC, activated protein C; PBS, phosphate-buffered saline.

**Figure 2 ijms-25-00370-f002:**
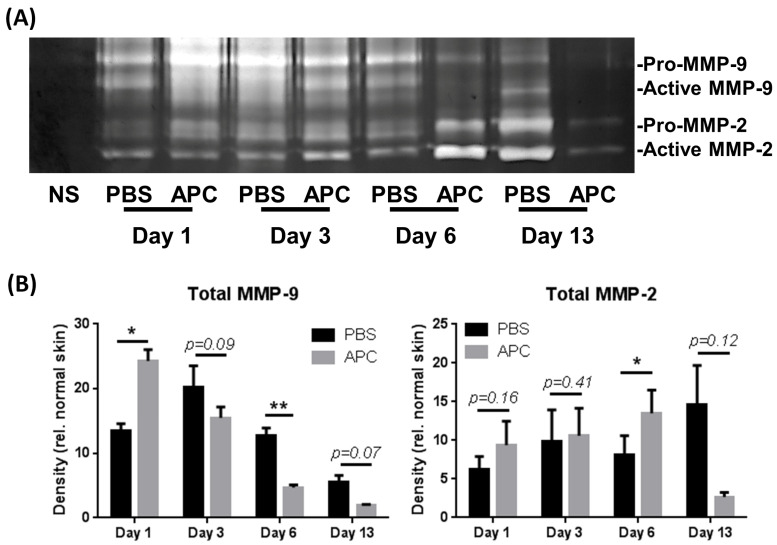
Temporal profile of MMP-2 and MMP-9 in healing wounds. Tissue homogenates were obtained from PBS- and APC-treated wounds on day 1, day 3, day 6, and day 13. (**A**) MMP-2 and -9 production was assessed using gelatin zymography. (**B**) Band densitometry was performed using ImageJ software V 1.8.0. Bars represent mean ± SD, *n* = 3. * *p* < 0.05, ** *p* < 0.01 vs. PBS-treated wounds by two-way ANOVA with Tukey’s post-hoc test. Abbreviations: APC, activated protein C; MMP, matrix metalloproteinase, PBS, phosphate-buffered saline.

**Figure 3 ijms-25-00370-f003:**
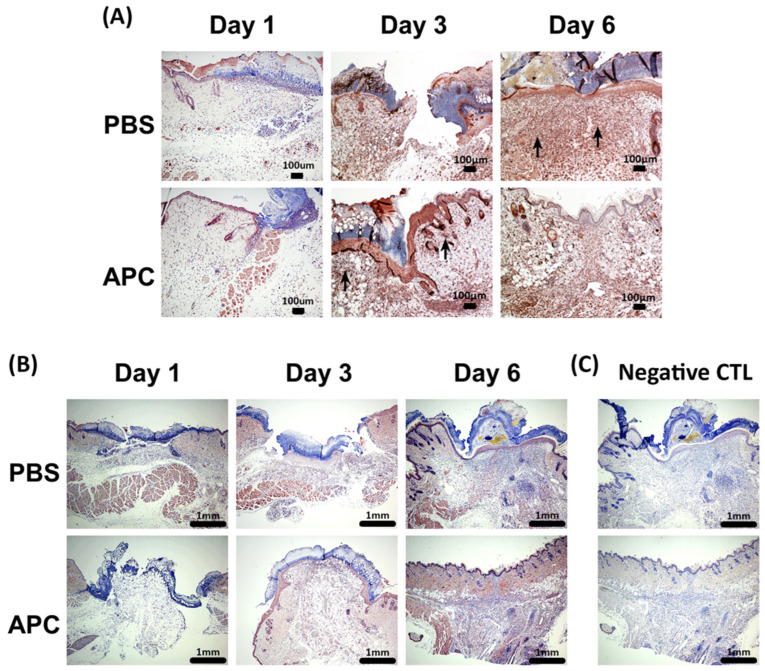
Immunohistochemical analysis of MMP-2 and MMP-9 in healing wounds. (**A**) Representative MMP-9 staining of PBS- and APC-treated wounds on day 1, day 3, and day 6 showed strong epithelial localization in APC-treated wounds on day 3 and on day 6 in PBS-treated wounds. (**B**) Representative MMP-2 staining of PBS- and APC-treated wounds on day 1, day 3, and day 6 showed an increased MMP-2 expression in both the epidermis and dermis in APC-treated wounds on day 6. (**C**) Negative control staining—rabbit IgG at the same dilution of MMP-2 and MMP-9. Scale bar = 1000 µm. Abbreviations: APC, activated protein C; MMP, matrix metalloproteinase, PBS, phosphate-buffered saline.

**Figure 4 ijms-25-00370-f004:**
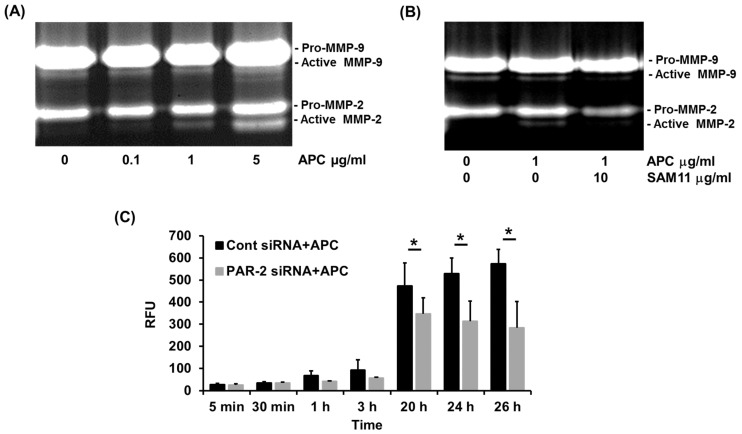
PAR-2 mediates APC-induced MMP-2 production by primary human keratinocytes. Cell supernatant was obtained from cultured primary human keratinocytes and MMP-2 and -9 production was assessed using gelatin zymography. (**A**) A dose-dependent increase in MMP-2 activation by 0–5 μg/mL of APC over 48 h. (**B**) Effect of 10 μg/mL of the PAR2-blocking antibody, SAM11, on APC-induced MMP-2 activation. (**C**) PAR-2 siRNA-transfected keratinocytes were treated with 1 μg/mL of APC for 48 h. MMP-2 within the cell supernatant was activated by the addition of 1 mM amino-phenyl mercuric acetate (APMA). The fluorescent intensity was measured at indicated time points. Bars represent mean ± SD, *n* = 4; * *p* < 0.05, by one-way ANOVA. Abbreviations: APC, activated protein C; MMP, matrix metalloproteinase, PAR, protease-activated receptor; RFU, relative fluorescent units; siRNA, small interfering ribonucleic acid.

**Figure 5 ijms-25-00370-f005:**
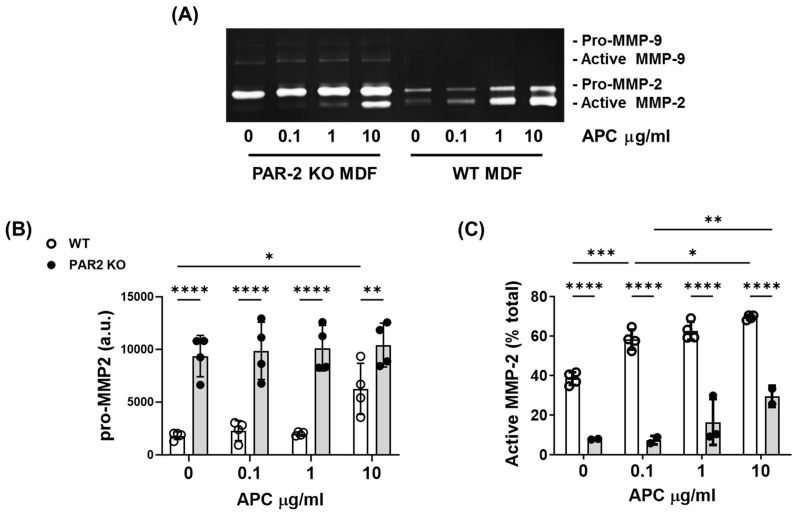
PAR-2 mediates APC-induced MMP-2 activation by primary murine dermal fibroblasts. Cell supernatant was obtained from cultured PAR-2 KO- and WT-mice dermal fibroblasts and treated with 0–10 μg/mL of APC for 48 h. (**A**) Representative gel of gelatin zymography, showing MMP-2 and -9 production. (**B**) Densitometric analysis of pro-MMP2. (**C**) The proportion (%) of active MMP-2 was analysed via band densitometry using ImageJ software. Bars represent mean ± SD, *n* = 4. * *p* < 0.05, ** *p* < 0.01, *** *p* < 0.001, **** *p* < 0.0001 by 2-way ANOVA with Tukey’s post-hoc test. Abbreviations: APC, activated protein C; a.u., arbitrary unit; MMP, matrix metalloproteinase, PAR, protease-activated receptor.

**Figure 6 ijms-25-00370-f006:**
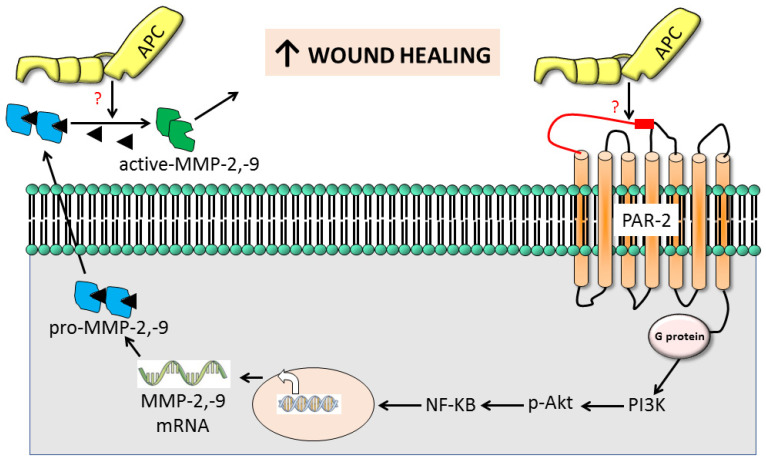
Proposed mechanism of PAR-2-induced gelatinolytic MMP activation in APC-stimulated wound healing. Based on data presented in this study, our previous studies, and other studies, we propose that APC-mediated PAR2 activation causes intracellular activation of AKT via PI3K, which phosphorylates IĸBα, leading to its proteasomal degradation, leaving NF-ĸB free to translocate into the nucleus to promote MMP-2 and MMP-9 expression. Released pro-active MMP-2 and MMP-9 are cleaved by APC through unknown mechanisms and promote cutaneous wound healing. ? indicates unknown mechanism (s).

## Data Availability

All data associated with this study are present in the paper or online Appendix A. Data supporting the findings of this study are available from the corresponding authors upon reasonable request.

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
