# Peer review of "Involvement of PAR-2 in the Induction of Cell-Specific Matrix Metalloproteinase-2 by Activated Protein C in Cutaneous Wound Healing"

_ijms, 2023, doi:10.3390/ijms25010370_

Round 1

Reviewer 1 Report

Comments and Suggestions for Authors

The manuscript is devoted to the study of the molecular mechanisms of wound regeneration, the role of C reactive protein and metalloproteinases in wound healing, and illustrates the staged nature of the process, which fits into today's ideas about the cellular mechanisms of healing.

The manuscript is understandable, relevant and well structured. Тhe manuscript is scientifically sound, the research methods are described in sufficient detail, which allows other researchers to repeat the work to test the hypothesis put forward by the authors.

The authors’ concept is relevant, especially since they use both morphology with immunohistochemistry and molecular studies of proteins, which are widely accepted today, as arguments in their hypothesis. The conclusions are consistent with the evidence and arguments presented. The presented work is consistent with the ethical standards accepted in Australia, as stated by the authors in the article.

The truth is that the very high number of links to your own publications is annoying. For a short time, the thought even appears whether the authors are the founders of this trend in the world.

Main Concerns:

It is necessary to reduce the number of self-citations (no more than 10-15% of references); this can be achieved both by reducing the number of self-citation references and by increasing the number of references to the works of other authors over the past 5 years.

The submitted manuscript contains excessive self-citation (about a third of all references).

Of the 51 publications in the list of references, the authors' articles are cited at least 14 times (references No. 2,11, 12, 13, 14, 17,18, 22, 23, 25, 44, 45, 46, 47). At the same time, there are only 10 references over the last 5 years to the works of other authors. According to the PubMed database, more than 4,500 publications upon request (matrix metalloproteinase AND wound healing), so there is a lot of choice.

Author Response

The manuscript is devoted to the study of the molecular mechanisms of wound regeneration, the role of C reactive protein and metalloproteinases in wound healing, and illustrates the staged nature of the process, which fits into today's ideas about the cellular mechanisms of healing.

The manuscript is understandable, relevant and well structured. Тhe manuscript is scientifically sound, the research methods are described in sufficient detail, which allows other researchers to repeat the work to test the hypothesis put forward by the authors.

The authors’ concept is relevant, especially since they use both morphology with immunohistochemistry and molecular studies of proteins, which are widely accepted today, as arguments in their hypothesis. The conclusions are consistent with the evidence and arguments presented. The presented work is consistent with the ethical standards accepted in Australia, as stated by the authors in the article.

The truth is that the very high number of links to your own publications is annoying. For a short time, the thought even appears whether the authors are the founders of this trend in the world.

Author Response: We would like to thank the reviewer for these comments.

Main Concerns:

It is necessary to reduce the number of self-citations (no more than 10-15% of references); this can be achieved both by reducing the number of self-citation references and by increasing the number of references to the works of other authors over the past 5 years.

The submitted manuscript contains excessive self-citation (about a third of all references).

Of the 51 publications in the list of references, the authors' articles are cited at least 14 times (references No. 2,11, 12, 13, 14, 17,18, 22, 23, 25, 44, 45, 46, 47). At the same time, there are only 10 references over the last 5 years to the works of other authors. According to the PubMed database, more than 4,500 publications upon request (matrix metalloproteinase AND wound healing), so there is a lot of choice.

Author Response: We agree with the reviewer. We have deleted the 11 self-citations (old references: 2, 13, 14, 21, 22, 25, 27, 44, 45, 46 and 47) and included 10 new recent references in the revised manuscript.

Reviewer 2 Report

Comments and Suggestions for Authors

Julovi et al. investigated the role of PAR-2 and APC in the context of wound healing. This is an interesting study, however, there are a couple of points authors need to clarify: 

1. Authors imply that APC prevent apoptosis and cite work by Xue et al. in support of their claim. However, Xue et al. work is very crude in the sense of elucidation of APC's role of apoptosis. Protein C is a GLA factor that can bind calcium, therefore what Xue et al. observed can simply be calcium chelation rather that true intracellular mechanism of apoptosis regulation. 

2. Are images presented in Figures 1 and 3 representative of the study? Have authors compared the wounds at timepoint 0 (right after wounding)? On images presented, the amount of muscular filaments is significantly more in PBS treated groups vs APC. This may suggest an observer bias of the study. Please elucidate this point. 

Author Response

Julovi et al. investigated the role of PAR-2 and APC in the context of wound healing. This is an interesting study, however, there are a couple of points authors need to clarify: 

  1. Authors imply that APC prevent apoptosis and cite work by Xue et al. in support of their claim. However, Xue et al. work is very crude in the sense of elucidation of APC's role of apoptosis. Protein C is a GLA factor that can bind calcium, therefore what Xue et al. observed can simply be calcium chelation rather that true intracellular mechanism of apoptosis regulation. 

Author Response: We thank the reviewer for this suggestion and agree that sequestration of calcium by the GLA region may partly contribute to the anti-apoptotic effect of APC. Nonetheless, direct anti-apoptotic mechanisms of APC are well described by numerous studies in different cells and systems. Some examples:

  • APC prevents neural and brain endothelial apoptosis by inhibiting tumor suppressor protein p53 and the proapoptotic protein Bax and promoting the antiapoptotic protein Bcl-4- this protective action of APC is mediated by both EPCR and PAR-1 receptors (Cheng et al Nat Med. 2003 Mar;9(3):338-42).
  • APC reduces NLRP3 inflammasome activation and subsequent apoptosis given before induction of myocardial or renal ischaemia–reperfusion injury in mice, the mechanism involves inhibition of PAR1 and mTORC1. (Nazir, Blood 2017)
  • APC modulates the mitochondrial apoptosis pathway via PAR-1 and EPCR to inhibit glucose-induced glomerular apoptosis in endothelial cells and podocytes. (Isermann et al, Nature Medicine, 13, 1349–1358 2007)
  • APC prevents neuronal apoptosis via nuclear translocation of apoptosis-inducing factor (AIF), induction of p53, inhibition of caspase-8 activation leading to caspase-3 - these effects of APC require PAR1 and PAR3 (Guo et al, Neuron, 41, 563-572, 2004)
  • APC downregulates the expression of TNF-related apoptosis inducing ligand (TRAIL) in endothelial cells, dependent on the activation of Egr-1/Erk1/2. The effect of APC was PAR-1– and spingosine-1-phosphate(S1P)-1-dependent (O’Brien et al, Arterioscl Thromb Vasc Biol, 2007).
  • APC prevents LPS-Induced apoptosis in endothelial cells via glycogen synthase kinase-3β (Luo et al, Med. Inflamm, 2012)

We have included a brief explanation of the above in the Introduction in page 2, lines 54 to 57.

  1. Are images presented in Figures 1 and 3 representative of the study? Have authors compared the wounds at timepoint 0 (right after wounding)? On images presented, the amount of muscular filaments is significantly more in PBS treated groups vs APC. This may suggest an observer bias of the study. Please elucidate this point. 

Author Response: In addition to Figures 1 and 3, we have presented macroscopic images of wounding on day 0, day 3, and day 6 (Supplemental Figure 1). The manuscript also includes high-magnification images of MMP2 expression on day 6 and MMP-9 expression on day 1 (Supplemental Figure 2). The number of muscular filaments is higher in the PBS-treated group in Figures 1 and 3B on day 1 and day 3, which might be characteristic of a delayed or unhealed wound due to the loss of cutaneous layers. A similar pattern of muscular filaments is observed on day 6.

On the other hand, sample collection, paraffin processing, and selections were performed blindly and randomly to prevent bias. We added the blind processing and random selection of samples in the Materials and Methods section, in lines 253 and 258-259, respectively.

Reviewer 3 Report

Comments and Suggestions for Authors

The manuscript by Julovi et al. entitled, "Involvement of PAR-2 in the induction of matrix metalloproteinase-2 by activated protein C in cutaneous wound healing" describes that PAR-2 is essential for APC-induced MMP-2 production and explains the underlying mechanisms by which APC promotes wound healing through PAR-2. After APC intervention, the murine skin wound healing and human skin keratinocytes were used to characterize the effects on MMP-2 and MMP-9 production. The contents of the manuscript can be of interest to the readers of Int. J. Mol. Sci. However, there are some issues in the manuscript that should be solved before recommended for publication.

1. In the section “Activated protein C accelerates murine cutaneous wound healing”. The histological analyses were performed on 1, 3, and 6. Were there longer experimental observations on day 13?

2. In Figures 1 and 3, it was suggested that the histological graph should be followed by the quantified analyses by Imag J software.

3. The parts of the article that need to be fine-tuned:

a.      Line 25, “APC stimulated murine cutaneous wound healing was associated with differential 0temporal production of MMP-2 and MMP-9,”

b.     In Figure 3A, the scale bar on days 3 and 6 should be in a standard format with day 1.

Comments on the Quality of English Language

Minor editing

Author Response

The manuscript by Julovi et al. entitled, "Involvement of PAR-2 in the induction of matrix metalloproteinase-2 by activated protein C in cutaneous wound healing" describes that PAR-2 is essential for APC-induced MMP-2 production and explains the underlying mechanisms by which APC promotes wound healing through PAR-2. After APC intervention, the murine skin wound healing and human skin keratinocytes were used to characterize the effects on MMP-2 and MMP-9 production. The contents of the manuscript can be of interest to the readers of Int. J. Mol. Sci. However, there are some issues in the manuscript that should be solved before recommended for publication.

Author Response: We would like to thank the reviewer for these comments.

  1. In the section “Activated protein C accelerates murine cutaneous wound healing”. The histological analyses were performed on 1, 3, and 6. Were there longer experimental observations on day 13?

Author Response: We have studied morphological changes on day 13 in our previous study. On day 13, in both control and in response to APC, PAR-2 KO mice exhibited thickened epithelial layers, a migrating epithelial tongue, and disorganized dermis. In contrast, WT mice showed a normal thin epithelial layer, a more mature dermis, and a lack of inflammatory cells when treated with APC (Ref 22).

Top of Form

  1. In Figures 1 and 3, it was suggested that the histological graph should be followed by the quantified analyses by Imag J software.

Author Response: We performed ImageJ analysis only on Figure 1 (histological images) to measure the wound gap. This was described in the Results and Materials and Methods sections, in lines 82 and 265 to 266, respectively.

  1. The parts of the article that need to be fine-tuned:
  2. Line 25, “APC stimulated murine cutaneous wound healing was associated with differential 0temporalproduction of MMP-2 and MMP-9,”

Author Response: Thank you for noticing the typing error. We have corrected line 25 in the revised manuscript by deleting the '0' and adding 'and' between 'differential' and 'temporal’ production.

  1. In Figure 3A, the scale bar on days 3 and 6 should be in a standard format with day 1.

Author Response: Thank you for the suggestions. We corrected the scale in the revised Figure 3A accordingly.

Reviewer 4 Report

Comments and Suggestions for Authors  

This is a well-done study following up on previous work by the authors and others on the role of PAR-2 in the induction of MMT-2 and -9 by activated protein C in wound healing. Two minor additions would improve the MS: First, describe the experiment at the beginning of the Results section rather than assuming the reader knows exactly what was done. Second, explain more clearly exactly what distinguishes this study from the previous one by the authors cited in ref. 17, in terms of the details of the proposed mechanism (Fig. 6) that are revealed in each study. Note that the word faint in line 129 is misspelled as feint.

Comments on the Quality of English Language

NONE

Author Response

We would like to thank the reviewer for these comments.

We corrected the spelling feint in the revised manuscript in line120.

Reviewer 5 Report

Comments and Suggestions for Authors

The manuscript „Involvement of PAR-2 in the induction of matrix metalloproteinase-2 by activated protein C in cutaneous wound healing”. I believe that this title asks for inclusion of MMP-9 and the phrase “cell-type specific”.

The authors should very clearly state the novel aspect of this research as the effect of APC in human keratinocytes was shown back in 2004. It was undoubtedly shown that APC induces MMP-2 in keratinocytes. So, the stronger accent should be put on PAR-2 as a mediator involved in APC-MMP-2 signaling.

My general impression is that this manuscript is nicely written and logically composed.

However, since the authors worked with only one si-RNA (and not two, which should be a golden standard), they must present efficacy of silencing performed.

The authors also mention PAR-1 KO animals, but I see no experiments performed in this  model of investigation.

Pretreatment with SAM11 should be specified.

Thank you.

Author Response

The manuscript „Involvement of PAR-2 in the induction of matrix metalloproteinase-2 by activated protein C in cutaneous wound healing”. I believe that this title asks for inclusion of MMP-9 and the phrase “cell-type specific”.

Author Response: Thank you for the suggestions. To elucidate the involvement of PAR-2 in MMP-2 and MMP-9 production by activated protein C (APC), although we used cultured neonatal foreskin keratinocytes with or without intact PAR-2 signaling and murine dermal fibroblasts from PAR-2 knock-out (KO) mice, we also used murine wounds tissue homogenates from PAR-2 KO mice, with the overall major robust finding being that PAR-2 is involved in MMP-2 induction by APC. We have changed the title to “Involvement of PAR-2 in the induction of cell-specific matrix metalloproteinase-2 by activated protein C in cutaneous wound healing

The authors should very clearly state the novel aspect of this research as the effect of APC in human keratinocytes was shown back in 2004. It was undoubtedly shown that APC induces MMP-2 in keratinocytes. So, the stronger accent should be put on PAR-2 as a mediator involved in APC-MMP-2 signaling.

Author Response: We completely agree with the reviewer, and anticipate that our current title and other descriptions including sentences in final discussion paragraph “A novel finding of this study is that PAR-2 is essential for APC-induced MMP-2 production……” and abstract “This study shows for the first time that PAR-2 is essential for APC-induced MMP-2 production."  accurately emphasise the novel role of PAR2.   

My general impression is that this manuscript is nicely written and logically composed.

 Authors Response: Thank you for these comments.

However, since the authors worked with only one si-RNA (and not two, which should be a golden standard), they must present efficacy of silencing performed.

 Authors Response: We have described the silencing efficacy and presented the efficacy rate with the representative western blot in Supplementary Figure S3 (reference 22). We agree with the reviewer that using two siRNAs is the gold standard for silencing studies. In addition to PAR-2 siRNA, we also employed a PAR-2-specific blocking antibody as a complementary approach to the siRNA experiments.

The authors also mention PAR-1 KO animals, but I see no experiments performed in this  model of investigation.

Authors Response: We apologise for the typing errors and deleted PAR-1 mice in lines 2 and 311 in the revised manuscript.

Pretreatment with SAM11 should be specified.

Author Response:  Cells were pre-treated with the PAR-2 blocking antibody SAM11 for 2 hours, and specified in Materials and Methods, line 269.